# Multi-Omics Analysis Reveals the Role of Sigma-1 Receptor in a Takotsubo-like Cardiomyopathy Model

**DOI:** 10.3390/biomedicines11102766

**Published:** 2023-10-12

**Authors:** Yi Liu, Qing Chen, Jian-Zheng Yang, Xiu-Wen Li, Li-Jian Chen, Kai-Kai Zhang, Jia-Li Liu, Jia-Hao Li, Clare Hsu, Long Chen, Jia-Hao Zeng, Qi Wang, Dong Zhao, Jing-Tao Xu

**Affiliations:** 1Guangzhou Key Laboratory of Forensic Multi-Omics for Precision Identification, School of Forensic Medicine, Southern Medical University, Guangzhou 510515, Chinawangqi1980@smu.edu.cn (Q.W.); 2Key Laboratory of Evidence Science, China University of Political Science and Law, Ministry of Education, Beijing 100088, China; 3Beijing Municipal Public Security Judicial Appraisal Center, Beijing 100142, China

**Keywords:** Sigmar1, Takotsubo syndrome, microbiome, metabolome, transcriptome

## Abstract

Takotsubo syndrome (TTS) is a stress-induced cardiomyopathy that presents with sudden onset of chest pain and dyspneic and cardiac dysfunction as a result of extreme physical or emotional stress. The sigma-1 receptor (Sigmar1) is a ligand-dependent molecular chaperone that is postulated to be involved in various processes related to cardiovascular disease. However, the role of Sigmar1 in TTS remains unresolved. In this study, we established a mouse model of TTS using wild-type and Sigmar1 knockout mice to investigate the involvement of Sigmar1 in TTS development. Our results revealed that Sigmar1 knockout exacerbated cardiac dysfunction, with a noticeable decrease in ejection fraction (EF) and fractional shortening (FS) compared to the wild-type model. In terms of the gut microbiome, we observed regulation of Firmicutes and Bacteroidetes ratios; suppression of probiotic Lactobacillus growth; and a rise in pathogenic bacterial species, such as Colidextribacter. Metabolomic and transcriptomic analyses further suggested that Sigmar1 plays a role in regulating tryptophan metabolism and several signaling pathways, including MAPK, HIF-1, calcium signaling, and apoptosis pathways, which may be crucial in TTS pathogenesis. These findings offer valuable insight into the function of Sigmar1 in TTS, and this receptor may represent a promising therapeutic target for TTS.

## 1. Introduction

Takotsubo syndrome (TTS), commonly referred to as “broken heart syndrome”, is a condition that induces symptoms that resemble those of heart disease, including chest pain and shortness of breath, in response to major emotional or physical stressors [1,2,3]. TTS is more commonly observed in female patients in clinical practice. A comment on the heightened risk of developing TTS in post-menopausal women is due to the lack of estrogen, a phenomenon commonly observed in clinical practice [4,5,6,7]. Studies have demonstrated a correlation between TTS and serious complications, including death, because of the development of cardiovascular complications such as cardiogenic shock and life-threatening arrhythmias [8,9,10]. Particularly in the context of the COVID-19 pandemic, accurate diagnosis of TTS has become more challenging. Cardiovascular involvement is relatively common in COVID-19 patients, and the incidence of TTS is significantly increased under conditions of pre-existing coronary endothelial dysfunction (CED) and stress. Episodes of TTS during this period could potentially be identified via acetylcholine (ACh) testing, although this hypothesis warrants further validation [11]. The mortality rate of TTS is similar to that of ST-elevation myocardial infarction (STEMI) [12]. The precise mechanism underlying the development of TTS is not yet fully understood but may be triggered as a result of excessive activation of the sympathetic nervous system in reaction to emotional or physical stressors, leading to an increase in plasma epinephrine and norepinephrine levels that cause damage to the heart [13,14]. A reference to the recent scientific statement from the Heart Failure Association of the European Society of Cardiology underlines the central role of catecholamines and the sympathetic nervous system in TTS [15]. Persisting or chronic hyperactivity of the sympathetic nervous system may influence the long-term outcomes of TTS. Admission hyperglycemia, a significant predictor of mortality, likely exacerbates sympathetic tone dysfunction and cardiac denervation in TTS patients [16]. Indeed, cardiac adrenergic receptors regulate cardiac function in both physiological and pathological states and influence the development of cardiac disease through various adaptive mechanisms and susceptibilities. Research has shown that an adrenergic receptor blocker could reduce the adverse recruitment of adrenergic receptors in chronic heart failure, suggesting that adrenergic receptors could be therapeutic targets for TTS, particularly in contexts with abnormal activation of the adrenergic system [17]. In clinical settings, a number of effective cancer therapeutics, known for their cardiotoxicity, have been associated with TTS. Case reports suggest that the adverse effects of these drugs, combined with the emotional distress and stress experienced during treatment, may precipitate TTS. Hence, it is crucial to elucidate the underlying causes of TTS in this context [18]. Although most current research on TTS has focused on its clinical characteristics, further investigation into the underlying mechanisms of TTS is needed to identify more effective and safer therapeutic approaches.

Sigmar1, also known as the sigma-1 receptor, is a molecular chaperone that performs a crucial function in both the physiological and pathological processes of the heart. Sigmar1 modulates ion channel activity and signaling pathways, resulting in the preservation of myocardial tissue [19]. Impairment of mitophagy has been observed in mice with Sigmar1 deficiency, suggesting that Sigmar1 may serve as a novel modulator of autophagy regulation [20]. Additionally, studies have demonstrated that Sigmar1 knockout in cardiac physiology leads to mitochondrial morphology and functional changes, resulting in cardiac dysfunction [21]. The involvement of Sigmar1 in cardiac pathophysiology has also been demonstrated in studies [19,22]. Nevertheless, the function of Sigmar1 in the pathogenesis of TTS remains unresolved.

Recent research indicates that the gut microbiome may be closely associated with cardiovascular function, as well as the fact that certain bacteria and their metabolites may facilitate the development of cardiovascular diseases such as coronary artery disease and heart failure. The gut microbiome has been suggested to have a role in influencing cardiovascular health, and several studies have shed light on the potential underlying mechanisms [23,24,25,26,27].

Metabolomics is a field of study that aims to characterize and quantify the metabolic profiles of biological systems, including those involved in the cardiovascular system. By analyzing changes in metabolite composition, metabolomics can provide insight into underlying metabolic pathways and mechanisms that regulate physiological processes and disease states. In humans, the metabolome is closely linked to heart health, with changes in metabolite levels serving as markers of heart injury or disease [28].

In this study, we used a multi-omics approach to investigate the role of Sigmar1 in an isoprenaline hydrochloride (ISO)-induced TTS-like cardiomyopathy mouse model. By analyzing the gut microbiome, serum metabolome, and heart transcriptome, we gained a deeper understanding of the mechanisms through which Sigmar1 may influence the development of TTS. Our findings revealed new research targets for investigating the pathological mechanisms underlying TTS and could facilitate the development of innovative therapeutic approaches for the treatment of TTS.

## 2. Materials and Methods

### 2.1. Animals

Mice were based on the C57BL/6J background, and females were chosen for experiments at 8–10 weeks of age. The Sigmar1 knockout (KO) mouse model was created by CRISPR/Cas-mediated genome engineering. We extracted genomic DNA from tails to confirm global Sigmar1 KO using wild-type mice as controls (Appendix A). Wild-type mice were purchased from the Experimental Animal Center of Southern Medical University (Guangzhou, China). The experiments were approved by the Ethics Committee of Southern Medical University (Ethical Committee Approval Code: SMUL2022091). Mice were allowed to acclimatize for one week and fed with standard laboratory food and water and reared under a 12 h light/dark cycle.

### 2.2. Mouse TTS Model

A mouse model of TTS was induced in our study by a single high-dose administration of ISO (200 mg/kg) [29]. This mouse TTS model has been reported to be a practical and valuable tool in research related to TTS. ISO (Solarbio, Beijing, China) was dissolved in phosphate-buffered saline (PBS, Solarbio, Beijing, China) and kept in darkness at 4 °C. We employed both wild-type mice and Sigmar1 knockout mice in our experiment. Each type was divided into two distinct groups: one group received a single intraperitoneal injection of ISO, while the other group was given PBS as a control. This resulted in four experimental groups, namely, WT_C (wild-type control), WT_ISO (wild-type with ISO treatment), KO_C (Sigmar1 knockout control), and KO_ISO (Sigmar1 knockout with ISO treatment), each consisting of 8 mice. After 24 h of drug treatment, the mice were all subjected to echocardiography and then sacrificed under anesthesia with 0.5% pentobarbital sodium by intraperitoneal injection. Serum and heart tissue were collected immediately and stored at −80 °C for future examination. (Figure 1A).

### 2.3. Echocardiography

Mice were anesthetized using 3% isoflurane for induction in a restraining box and 1% isoflurane for maintenance by mask [30]. We conducted transthoracic echocardiography using a Vevo2100 ultra-high frequency imaging system equipped with an 18–38 MHz linear array transducer (Visualsonics, Toronto, ON, Canada). After obtaining the echocardiography images, the left-ventricular end-diastolic and -systolic anterior wall thickness (LVAWd and LVAWs), left-ventricular end-diastolic and -systolic posterior wall thickness (LVPWd and LVPWs), and left-ventricular end-diastolic and -systolic internal diameter (LVIDd and LVIDs) were measured over three cardiac cycles, and averages were calculated. Finally, we calculated the ejection fraction (EF, %) and fractional shortening (FS, %) percentage using formulas that were previously described [31,32].

### 2.4. Fecal Microbiota Analysis by 16S rRNA Sequencing

One day after the establishment of the TTS mouse model, we collected fecal samples and preserved them at −80 °C for future use [33,34]. After the fecal samples were crushed, DNA extraction and PCR amplification were performed in accordance with the kit instructions. Then, to assess the DNA concentration, we utilized a NanoDrop2000 (Thermo Fisher Scientific, Waltham, MA, USA), while the quality of the DNA was analyzed by 1% agarose gel electrophoresis. Primers 228F and 806R were used to amplify the bacterial 16S rRNA gene in the variable region of V3–V4. Following the standard protocols from Majorbio Bio-Pharm Technology Co., Ltd. (Shanghai, China), the amplification products were used to construct sequence libraries on an Illumina MiSeq platform (Illumina, San Diego, CA, USA). We clustered the operational taxonomic units (OTUs) with a 97% similarity cutoff using UPARSE (version 7.1 http://drive5.com/uparse/ (accessed on 15 September 2021)) according to standard procedures. A-Diversity indices were calculated by Mothur (version 1.30.2), and principal coordinate analysis (PcoA) was conducted using R software (version 3.3.1). Bar charts at the phylum and genus levels were generated to analyze the gut microbial community composition of the samples. The Wilcoxon rank-sum test was used to identify differential bacteria with an FDR-corrected post hoc test. The linear discriminate analysis effect size (LefSe) was carried out to determine the communities or species significantly influenced by sample division. All analyses were performed on the Majorbio cloud platform (https://cloud.majorbio.com (accessed on 15 September 2021)).

### 2.5. Metabolic Analysis of Serum by LC–MS/MS

We extracted metabolites from serum samples obtained from the mice [34,35], and then we conducted LC–MS analysis utilizing a UHPLC-Q Exactive system (Thermo Fisher Scientific, Waltham, MA, USA). After mass spectrometry detection was completed, we imported the raw data of LC–MS into Progenesis QI (Version 3.0, Waters Corporation, Milford, MA, USA) software. Additionally, to search and identify metabolites, information from MS and MS/MS spectra was matched with the following databases: HMDB (http://www.hmdb.ca/ (accessed on 20 September 2021)), Metlin (https://metlin.scripps.edu/ (accessed on 20 September 2021)), and Majorbio. Differentially abundant metabolites were screened with a standard *p* value < 0.05 and a VIP (variable importance in projection) > 1, which was performed under the Wilcoxon test. The R package ropls (Version 1.6.2) was used for orthogonal least partial squares discriminant analysis (OPLS-DA) and for mapping the pathways through enrichment of the differentially abundant metabolites between two groups, which were based on the KEGG database (http://www.genome.jp/kegg/ (accessed on 20 September 2021)). Python packages (Version 1.11.3, https://docs.scipy.org/doc/scipy/ (accessed on 20 September 2021)) were used for statistically significant enrichment pathway analysis. We analyzed the data using the Majorbio cloud platform (https://cloud.majorbio.com (accessed on 20 September 2021)).

### 2.6. Transcriptome Analysis of Heart Tissue

We utilized TRIzol reagent to extract total RNA from the left heart tissue, and then we removed genomic DNA using DNase I. We assessed the RNA quality using a 2100 Bioanalyzer (Agilent) and quantified it using an ND-2000 (NanoDrop Technologies, Wilmington, DE, USA). We prepared 5 μg of total RNA using an RNA sample preparation kit (Illumina, San Diego, CA, USA). The extracted RNA was used for the construction of the cDNA library. Then, the cDNA amplified by PCR was sequenced by an Illumina HiSeq X Ten (Major Biotechnology Company, Shanghai, China). The raw reads were filtered for cleaning, and then unique consensus sequences were constructed by Trinity [36]. We used the Majorbio cloud platform (https://cloud.majorbio.com (accessed on 26 September 2021)) to analyze the data, screened all the differentially expressed genes (DEGs) between every two groups, and then analyzed the pathway enrichment based on KEGG. The DEGs were identified according to *p*-adjust < 0.05 and log2FC ≥ 1.

### 2.7. Correlation Analysis

We used the OmicStudio tools available at https://www.omicstudio.cn (accessed on 10 October 2021) to perform correlation estimates. The analysis was based on R version 3.6.3 (29 February 2020) and the Pearson calculation method. The results are reported using a clustering correlation heatmap.

### 2.8. Statistical Analysis

For data sets conforming to a normal distribution, differences between the two groups were assessed using a two-tailed Student’ s *t*-test or Welch’ s *t*-test and presented as the mean ± SEM (standard error of mean). For data sets not conforming to a normal distribution, the Wilcoxon test was applied. Statistical results were regarded as significant when *p* < 0.05. GraphPad Prism (Version 8.0.2; GraphPad Prism Software) was used to perform statistical analysis.

## 3. Results

### 3.1. ISO Led to TTS-like Cardiac Dysfunction

The cardiac damage induced by ISO was investigated by establishing a model of TTS in wild-type mice. Mice treated with a high dose of ISO exhibited a decrease in both EF and FS. Moreover, a balloon-like deformation of the left ventricle was observed in the B-mode image (Figure 1B–D). The results indicated the successful establishment of a mouse model of TTS induced by ISO, as evidenced by significant cardiac dysfunction.

### 3.2. TTS Induced by ISO Led to Gut Microbial Dysbiosis

To further investigate alterations in the gut microbiome linked to TTS, fecal samples from the wild-type control (WT_C) and ISO-treated (WT_ISO) groups were collected for 16S rRNA sequencing. The results showed that after the administration of ISO, the Shannon index increased and the Simpson index decreased (Figure 2A,B), both suggesting that the diversity of the gut microbiota was increased. For β-diversity, the PCoA results indicated that individuals in the WT_C and WT_ISO groups were clearly distinct from each other (Figure 2C). Distinct changes in microbial composition were observed in the gut microbiome between the two groups. The bar plot showed that Firmicutes and Bacteroidetes were the dominant phyla with distinct changes after ISO administration (Figure 2D). In particular, the proportion of Firmicutes decreased from 81.89% to 48.27%, whereas the proportion of Bacteroidetes increased from 10.51% to 44.70%. At the genus level, the principal bacteria were Lactobacillus, norank_f_Muribaculaceae, Lachnoclostridium, and Enterorhabdus (Figure 2E). A significant decrease in the abundance of Lactobacillus and a significant increase in the abundance of Colidextribacter were observed. In addition, the proportions of norank_f_Muribaculaceae, Alloprevotella, Rikenellaceae_RC9_gut_group, Faecalibaculum, Dubosiella, Bacteroides, and many other bacteria were largely increased in the WT_ISO group (Figure 2F–H). The results of LEfSe analysis revealed that Firmicutes was the major phylum that differed between the two groups, whereas NK4a214_group, Colidextribacter, and Ruminococcus were the major genera that differed. The relationship between the gut microbiome and cardiac injury was investigated by correlation analysis between the cardiac dysfunction indices and the top 40 bacterial genera (see heatmap, Figure 2I). Colidextribacter, Bacteroides and Rikenellaceae_RC9_gut_group correlated inversely with EF and FS, whereas probiotic Lactobacillus had an opposite relationship.

### 3.3. TTS Induced by ISO Led to Serum Metabolic Disturbance

Changes in the metabolome associated with TTS were investigated by comparing the metabolomes of the WT-C and WT_ISO groups. As shown in the PLS-DA score plot, the samples of the two groups exhibited a distinct division (Figure 3A). During the analysis process, we excluded an abnormal sample in the WT-C group that might affect the results. A total of 193 differentially abundant metabolites were identified, including 135 upregulated and 58 downregulated metabolites after administration of ISO (Appendix A) (blue represents upregulated and green represents downregulated metabolites in Figure 3B). According to the VIP value, a heatmap was generated to reflect significant differences in metabolites between the two groups (Figure 3C). We found that butyric acid, tauroursodexycholic acid, and taurocholic acid decreased in the WT_ISO group. These metabolites are involved in butanoate metabolism and taurine and hypotaurine metabolism. Based on the KEGG enrichment analysis, 48 pathways were enriched (Appendix A), including tryptophan metabolism and bile secretion (Figure 3D). Next, we selected the top 40 metabolites based on VIP values to conduct correlation analysis with the cardiac dysfunction indices (Figure 3E). The results indicated that the concentrations of butyric acid, tauroursodexycholic acid, and taurocholic acid were significantly positively correlated with EF and FS, which suggested that butyric acid, tauroursodexycholic acid, and taurocholic acid are protective factors against cardiac dysfunction. A correlation analysis was conducted between the top 40 metabolites and the top 40 bacterial genera in terms of abundance, and there were close relationships between metabolites and bacteria, as shown in the heatmap (Appendix A).

### 3.4. TTS Induced by ISO Led to Cardiac Transcriptome Variation

Then, we conducted RNA sequencing on the left ventricles of mice. Fifty-three upregulated and 87 downregulated genes between the WT_C and WT_ISO groups were identified (red, upregulated; green, downregulated; Figure 4A). A significant increase in Sigmar1 mRNA expression was observed in the WT_ISO group (Figure 4B). The details of differential gene expression are shown in the heatmap for each sample, which clearly shows hierarchical clustering (Figure 4C). KEGG enrichment analysis was used to determine the pathways regulated by DEGs. There were 149 enriched pathways in total (Appendix A), including HIF-1, MAPK, p53, calcium signaling, and apoptosis pathways (Figure 4D). These enriched pathways may be closely related to ISO-induced cardiac impacts in TTS.

### 3.5. Sigmar1 Knockout Resulted in Cardiac Dysfunction

Previous studies have demonstrated the advantageous impacts of Sigmar1 on cardiac function and have identified this receptor as a potential target for the prevention and treatment of cardiovascular disease [22,23]. We explored the difference between the Sigmar1 KO mice (KO_C) and wild-type mice (WT_C) groups as the baseline in the following experiments. As Figure 5A shows, damaging alterations were detected in the heart after Sigmar1 KO. Moreover, we observed decreases in both EF (Figure 5B) and FS (Figure 5C), indicating that Sigmar1 KO led to cardiac dysfunction.

### 3.6. Sigmar1 Knockout Resulted in Alterations in the Gut Microbial Composition, Serum Metabolites, and Cardiac Transcriptome

The impact of Sigmar1 was further investigated by performing 16S rRNA sequencing of mouse fecal samples, serum metabolomics, and cardiac transcriptomics of the WT_C and KO_C groups. The gut microbial diversity and richness were increased significantly in mice, as reflected by the Shannon and Simpson indices and the Ace and Sobs indices, respectively (Figure 6A–D). PCoA revealed that samples were separated well (Figure 6E), which demonstrated a significant difference between the two groups. Expanding the analysis of microbial compositions, we found that the microbial community compositions were clearly different at both the phylum and genus levels. The abundance of Firmicutes decreased from 81.89% to 68.14% and that of Bacteroidetes increased from 10.51% to 25.14% (Figure 6F). The magnitude of variations in these two predominant phyla was smaller when compared with the corresponding variations in the WT_ISO group. In addition, Lactobacillus, norank_f_Muribaculaceae, Lachnoclostridium, Enterorhabdus, and Alloprevotella were still the most enriched genera (Figure 6G), even though the ratio of Lachnospiraceae was clearly increased after Sigmar1 KO, becoming the fourth most enriched bacterial genus in the KO_C group. Moreover, a significant reduction in Lactobacillus abundance and an increase in Colidextribacter abundance were detected, and the abundance of norank_f_Muribaculaceae, Lachnospiraceae_NK4A136_group, Alloprevotella, Dubosiella, and Bacteroides increased markedly in the KO_C group (Figure 6H–J).

For metabolomics, a significant difference was observed between the WT_C and KO_C groups. A PLS-DA score plot reflected good separation (Figure 7A), which indicated that Sigmar1 KO evidently affected the mouse metabolome. In our study, Sigmar1 KO resulted in 113 upregulated metabolites and 50 downregulated metabolites (Appendix A) (blue, upregulated; green, downregulated; Figure 7B). The generated heatmap demonstrates the alterations in each differentially abundant metabolites concentration among the samples (top 30 VIP values), and there was distinct clustering between the two groups (Figure 7C). Tyramine glucuronide participates in multiple metabolic signaling pathways, such as pentose and glucuronate interconversions, bile secretion, ascorbate and aldarate metabolism, and its concentration declined in the KO_C group. In addition, 64 pathways were enriched by KEGG analysis (Appendix A), including the mTOR signaling pathway, apoptosis pathway and tryptophan metabolism, alpha-linolenic acid metabolism, and arachidonic acid metabolism (Figure 7D).

For transcriptomics analysis, 31 genes were upregulated, whereas 80 genes were downregulated, which were mapped in the MA plot (red represents upregulated and green represents downregulated; Figure 7E). The expression of these 126 DEGs was demonstrated in the heatmap (Figure 7F), which showed clear variation between the two groups. KEGG enrichment analysis identified 135 pathways (Appendix A), including the mTOR, calcium, and HIF-1 signaling pathways (Figure 7G). The mTOR pathway is closely associated with inflammation, apoptosis, and cell growth [37], and necrosis and apoptosis of cardiomyocytes can be induced by the mTOR pathway [38,39].

These results suggest that Sigmar1 KO disrupts the balance of the gut microbiome, leading to alterations in serum metabolites and changes in the cardiac transcriptome.

### 3.7. Sigmar1 Knockout Aggravated Cardiac Dysfunction in TTS Mice

We next compared the cardiac function indices between the WT_ISO and KO_ISO groups to investigate the role of Sigmar1 in TTS induced by ISO. The percentages of EF and FS were lower in the KO_ISO group than in the WT_ISO group (Figure 8A–C). These findings suggest that the combined effect of Sigmar1 deficiency and ISO administration leads to exacerbated cardiac injury.

### 3.8. Sigmar1 Knockout Aggravated Gut Microbial Dysbiosis in TTS Mice

The impact of Sigmar1 on TTS in the gut microbiome was further explored by conducting 16S rRNA sequencing, and the results suggested a significant difference between the WT_ISO and KO_ISO groups. No significant differences were observed regarding the diversity and richness of the gut microbiota (Appendix A). In contrast, PCoA showed that the samples were completely separated between the two groups (Figure 8D). Next, the bar chart of the microbial community composition showed changes in the percentages of Firmicutes and Bacteroidetes at the phylum level. Firmicutes accounted for 74.37%, and Bacteroidetes accounted for 18.64% in the KO_ISO group (Figure 8E). At the genus level, sharp increases in Allobaculum and norank_f_norank_o_Clostridia_UCG-014 abundances were observed (Figure 8F). Variations in the top 40 enriched bacterial genera were observed by heatmap analysis (Figure 8G). Furthermore, comparison at the genus level revealed a significant difference among those enriched bacteria. The abundance of Lactobacillus markedly decreased and that of Colidextribacter and Dubosiella Bacteroides clearly increased in the KO_ISO group (Figure 9A–E). LEfSe was carried out to identify the differential bacterium that made the strongest contribution to the classification, of which the top 3 were norank_f_Oscillosiraceae, unclassified_f_Oscillosiraceae, and Colidextribacter (Figure 9F). Correlation analysis between gut microbiota and cardiac impairment indices (Figure 9G) showed a statistically significant positive association between the abundance of Lactobacillus and ejection fraction (EF) and a statistically significant negative association between the abundance of Collidextribacter and both EF and fractional shortening (FS). These results indicate that perturbation of the gut microbiome is linked to the more severe cardiac damage observed in the Sigmar1 KO mice subjected to TTS by ISO induction.

### 3.9. Sigmar1 Knockout Aggravated Serum Metabolic Disturbance in TTS Mice

We continued to study serum metabolomics to explore the impact of Sigmar1 on TTS. The PLS-DA score plot revealed clear differences in metabolomics between the WT_ISO and KO_ISO groups (Figure 10A). There were 299 differentially abundant metabolites, including 123 upregulated and 176 downregulated metabolites in the KO_ISO group (Appendix A) (blue represents upregulated and green represents downregulated in the volcano plot; Figure 10B). The clustering heatmap (Figure 10C) showed metabolic variations and metabolite VIP values in detail. The results of KEGG enrichment analysis (Figure 10D) revealed that there was a notable enrichment of pathways associated with EGFR tyrosine kinase inhibitor resistance, parathyroid hormone synthesis, secretion and action, adrenergic signaling in cardiomyocytes, and tryptophan metabolism, which may contribute to cardiomyopathy. A total of 124 additional pathways are listed in Appendix A.

### 3.10. Sigmar1 Knockout Changed the Cardiac Transcriptome in TTS Mice

Finally, cardiac transcriptomics data were compared to investigate the mechanism of Sigmar1 regulation in TTS mice. Eighty-nine genes were significantly altered in the KO_ISO group, including 37 upregulated and 52 downregulated genes, when compared with the gene expression of the WT_ISO group (Figure 10E). All DEGs were plotted in the heatmap with clear clustering between the two groups (Figure 10F). Then, the pathways in which the DEGs were involved were identified by KEGG enrichment analysis, and some of those pathways (such as MAPK, HIF-1, p53, calcium signaling, and apoptosis pathways) are shown in the bar chart (Figure 10G), whereas the remaining pathways are listed in Appendix A. These pathways were also enriched in the comparison between the WT_C and WT_ISO groups, indicating that these pathways define the role of Sigmar1 in TTS.

## 4. Discussion

This study showed that ISO induces TTS-like cardiac dysfunction in mice and alterations in the gut microbiota, serum metabolites, and heart transcriptome. In particular, an increase in the growth of pathogenic bacteria and the suppression of probiotic bacteria in the ISO-treated group were observed. Additionally, Sigmar1 KO exacerbated cardiac damage in TTS and was associated with worse cardiac function. Analysis of the metabolome and transcriptome revealed regulatory changes in tryptophan metabolism and the MAPK, HIF-1, calcium signaling, and apoptosis pathways. These observations shed light on the role of Sigmar1 in TTS.

TTS is a stress-induced cardiomyopathy that is triggered by external events. ISO has been observed to induce damage resembling TTS in an animal model [31]. We chose female mice as subjects for our experiment to better model the clinical observation that TTS is more commonly observed in female patients. In this study, the results of the cardiac function examination supported the successful establishment of TTS. The relative abundance of Firmicutes and Bacteroidetes in the gut microbiota has been reported to be an indicator of microbial balance [40]. Our results showed that gut microbial dysbiosis occurred in the TTS model and that the ratios of Firmicutes and Bacteroidetes were significantly altered. Lactobacillus plays a critical role in immunoregulation. Some proteinaceous molecules with immunomodulatory activities from Lactobacillus have been identified to inhibit cell apoptosis induced under proinflammatory conditions [41]. Colidextribacter has been linked to negative impacts on human health, including the promotion of oxidative stress in human cells and the impairment of gut barrier function [42]. The abundance of Colidextribacter has also been found to increase in association with damage to intestinal permeability [43]. Therefore, the TTS model may be associated with impaired intestinal function, leading to harmful substances entering the bloodstream and affecting the heart, potentially because of Lactobacillus downregulation and Colidextribacter upregulation.

Metabolomics is a powerful analytical technique that provides insights into the metabolic pathways and mechanisms that regulate various physiological processes and disease states. In this study, we identified differentially abundant metabolites in the tryptophan metabolism and taurine and hypotaurine metabolism pathways between the WT_C and WT_ISO groups. Tryptophan metabolism and taurine and hypotaurine metabolism are tightly related to cardiovascular disease [44]. Taurine is a necessary substance for the generation of bile acids, which can decrease the expression of pro-inflammatory cytokines in the aorta [45]. Tryptophan metabolism is regulated by derivatives of gut microorganisms and is closely related to heart failure, coronary artery disease, and arrhythmia [46]. Butyric acid levels were found to be significantly decreased in the WT_ISO group in this study. Previous studies have shown the importance of butyric acid in several biological processes, such as maintaining gut health, and that supplementation with butyric acid can protect against cardiac dysfunction [47,48,49]. Although the mechanisms underlying TTS-induced myocardial injury are not fully understood, our analysis of cardiac transcriptomics identified several key pathways that were altered in response to ISO treatment, which may be relevant to the development of TTS. These pathways include the calcium, PI3K-AKT, and MAPK signaling pathways. These findings may offer novel insight into the damage mechanisms of TTS and warrant further investigation.

Sigmar1 is a molecular chaperone protein with inter-organelle signaling capabilities that has been shown to have cytoprotective and cardioprotective effects [50]. Our findings showed that Sigmar1 deficiency resulted in dysregulation of the gut microbiome, including suppression of the growth of the probiotic bacterium Lactobacillus and promotion of the pathogenic bacterium Colidextribacter. The alterations observed in the gut microbiome may play a role in the cytoprotective and cardioprotective effects of Sigmar1. We also observed perturbations in the metabolome of mice with Sigmar1 deficiency. Further analysis revealed that two of the pathways most significantly impacted by these changes were the mTOR signaling and apoptosis pathways. Sigmar1 has been demonstrated to modulate ion channel activity and signaling pathways, which can affect cardiac function [21]. Our findings from both metabolomics and transcriptomics analyses indicated that Sigmar1 deficiency leads to significant changes in the calcium signaling pathway.

To explore the potential involvement of Sigmar1 in the development of TTS induced by ISO, we conducted a multi-omics analysis by comparing the WT_ISO and KO_ISO groups. We observed gut microbial dysbiosis in the KO_ISO group, including an alteration in the ratios of Firmicutes and Bacteroidetes; a decrease in the abundance of Lactobacillus; and an increase in Colidextribacter, Dubosiella, and Bacteroides, which are strongly associated with cardiac function [51]. Correlation analysis revealed that the abundance of Lactobacillus was positively correlated with indicators of cardiac function, whereas the abundance of Colidextribacter had a negative correlation, indicating that these bacteria have a beneficial or harmful effect linked to cardiac function.

Furthermore, we observed that Sigmar1 knockout disturbed serum metabolites in TTS mice. Through an enrichment analysis of differentially abundant metabolites using the KEGG pathway database, we identified several important regulatory pathways involving Sigmar1, including tryptophan metabolism, taurine and hypotaurine metabolism, and arachidonic acid metabolism. The metabolism of arachidonic acid has been associated with the development of cardiovascular diseases, and its involvement in the cardiovascular system is postulated to have a beneficial effect [52,53]. The liberation of arachidonic acid from inflammatory cells has also been observed during the inflammatory process [51].

Transcriptome analysis also identified significant differences in gene expression between the WT_ISO and KO_ISO groups, with altered regulation of important pathways such as HIF1, MAPK, and p53 signaling pathways. HIF1 signaling has a crucial function in cardiovascular regulation because it modulates oxygen homeostasis and vascular remodeling, thereby protecting against ischemic heart disease [54]. Additionally, HIF1 signaling has been shown to play a role in the development of atherosclerosis through its effects on vascular smooth muscle cells and macrophages [55]. The MAPK signaling pathway participates in a variety of cellular processes, including cell growth, differentiation, and apoptosis. Dysregulation of this pathway has been implicated in various cardiovascular diseases, such as hypertension and heart failure [56,57,58]. The p53 signaling pathway is also closely linked to cardiovascular diseases, including TTS. p53 is a transcription factor that plays a critical role in regulating cell cycle progression and apoptosis, and its activation can lead to cardiomyocyte death [59,60,61,62]. Our results revealed enrichment in several immune-system-related signaling pathways. Notably, localized myocardial inflammatory changes were observed in both human post-mortem hearts and rats with TTS-like cardiomyopathy. This underscores the pivotal role that inflammation plays in TTS [63,64]. Certain natural bioactives, such as quercetin and polydatin, have been identified as potential agents for mitigating the inflammatory response. These substances may therefore be considered viable candidates for reducing the inflammatory response in TTS, potentially alleviating the pathological processes underlying this condition [65,66]. A previous study showed a significant improvement in sympathetic nerve function during a 12 month follow-up in TTS patients treated with α-lipoic acid. This finding underscores the potential therapeutic effects of anti-oxidative and anti-inflammatory drugs in these patients [67]. Dysregulation of these pathways may therefore contribute to the pathological processes underlying TTS.

## 5. Conclusions

We used multi-omics analysis to examine the effects of Sigmar1 on ISO-induced TTS-like cardiomyopathy in mice. We found that Sigmar1 deficiency caused dysregulation of the gut microbiome, changed the levels of serum metabolites, and exacerbated cardiac damage in response to ISO treatment. These findings suggest that Sigmar1 plays a protective role in TTS and may be a potential therapeutic target for the treatment of this stress-induced cardiovascular disorder.

## Figures and Tables

**Figure 1 biomedicines-11-02766-f001:**
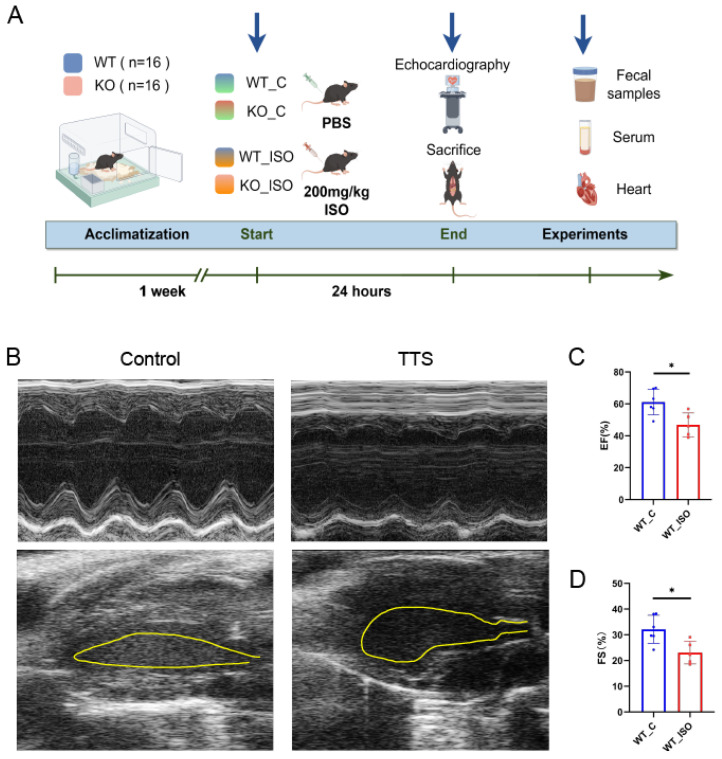
ISO induced TTS-like cardiac damage in WT mice. (**A**) Protocol scheme of the experiment. (**B**) Representative echocardiography images. (**C**,**D**) Indices of cardiac function: EF, FS (%) in mice. *n* = 5–6. Student’s *t*-test was used for statistical analysis. * *p* < 0.05.

**Figure 2 biomedicines-11-02766-f002:**
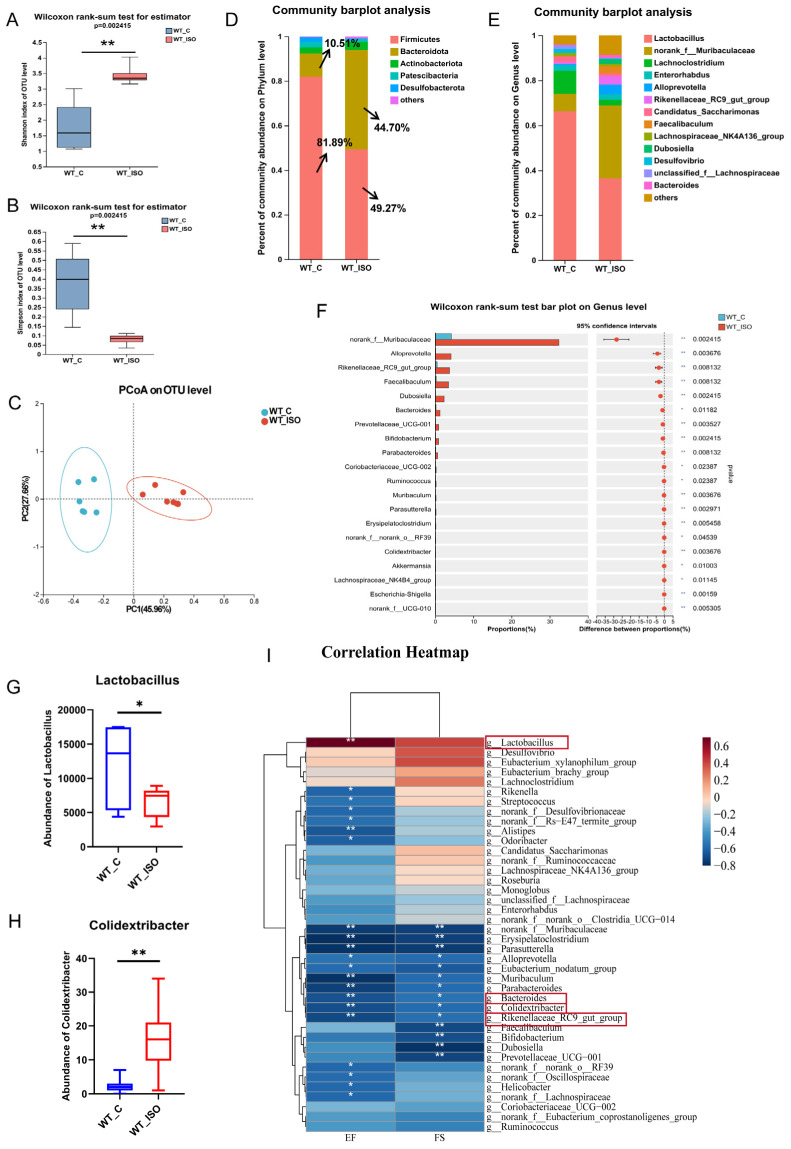
Gut microbial alteration in TTS mice. (**A**,**B**) Shannon index and Simpson index of α-diversity analysis. (**C**) PCoA plot of the gut microbiome. (**D**,**E**) The composition of the gut microbiome at the phylum and genus levels. (**F**–**H**) Significance test of differential bacteria at the genus level using the Wilcoxon rank-sum test, followed by the FDR-corrected post hoc test. * *p* < 0.05, ** *p* < 0.01. (**I**) Correlation analysis between differential bacteria (top 40 at the genus level) and indices of cardiac dysfunction. Red represents a positive relation, while blue represents a negative relation. The depth of color corresponds to the degree of connection.

**Figure 3 biomedicines-11-02766-f003:**
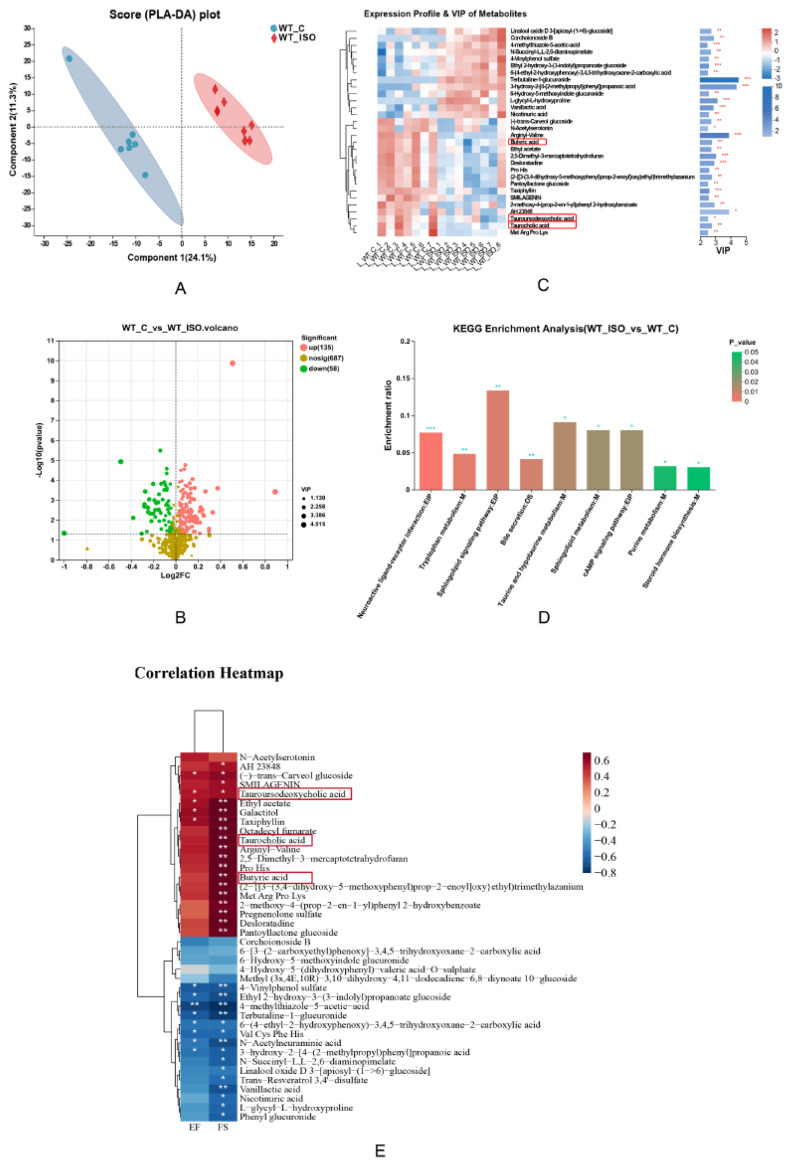
Metabolite variation in TTS mice. (**A**) PLS-DA score plot between the WT_C and WT_ISO groups. (**B**) Volcano plot of differentially abundant metabolites in mouse serum (*n* = 7 in the WT_C group, *n* = 8 in the WT_ISO group). (**C**) Heatmap showing the expression profile and VIP value of metabolites. Butyric acid, tauroursodeoxycholic acid, and taurocholic acid were downregulated in TTS mice. The VIP value was based on OPLS-DA. * *p* < 0.05, ** *p* < 0.01, *** *p* < 0.001. (**D**) KEGG enrichment analysis using the hypergeometric distribution algorithm. (**E**) Correlation analysis between differentially abundant metabolites (top 40 VIP) and indices of cardiac dysfunction.

**Figure 4 biomedicines-11-02766-f004:**
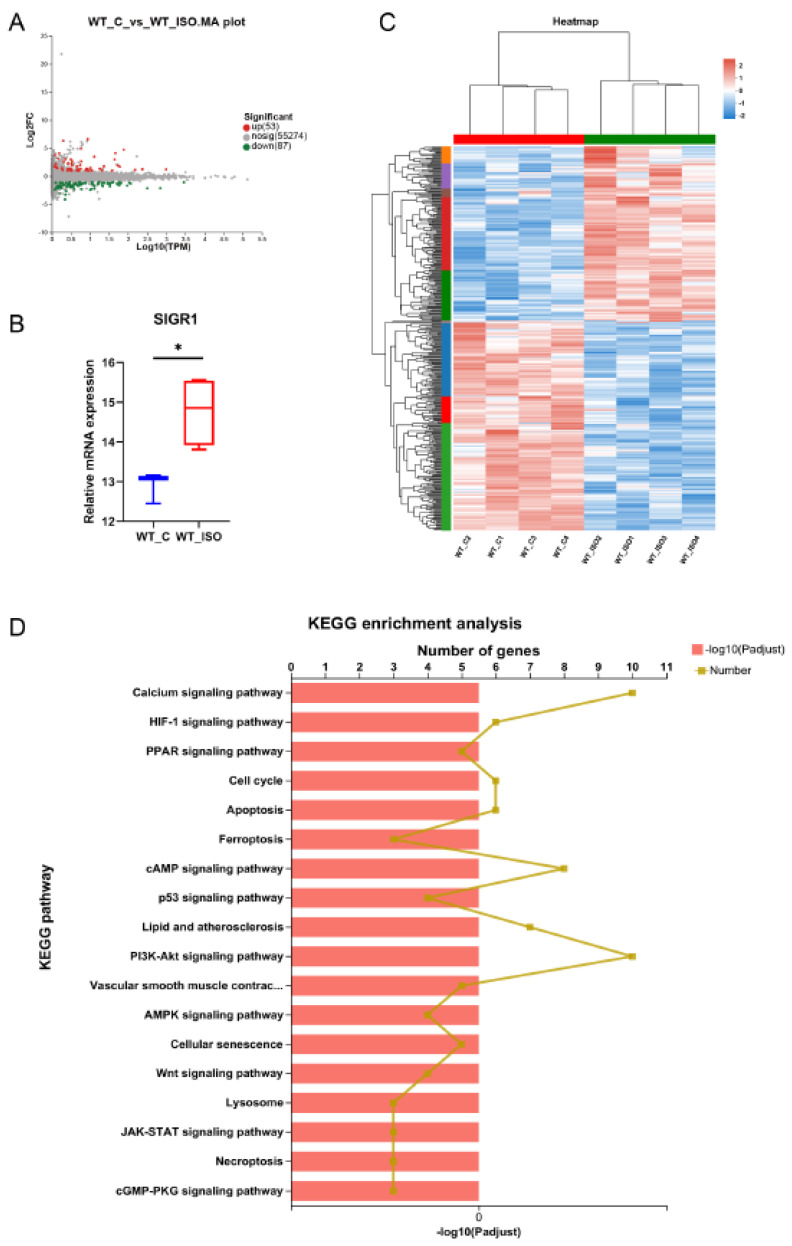
Transcriptomic variation in TTS mice. (**A**) MA plot of the differentially expressed genes (DEGs) between the WT_C and WT_ISO groups. (**B**) Relative mRNA expression of Sigmar1 was upregulated in TTS mice. *p*-value was from Student’s *t*-test. * *p* < 0.05. (**C**) The expression of all differentially expressed genes is shown in the heatmap. There was clear hierarchical clustering of samples between groups. (**D**) Specific potential pathways were selected for illustration in the bar chart of the KEGG enrichment analysis.

**Figure 5 biomedicines-11-02766-f005:**
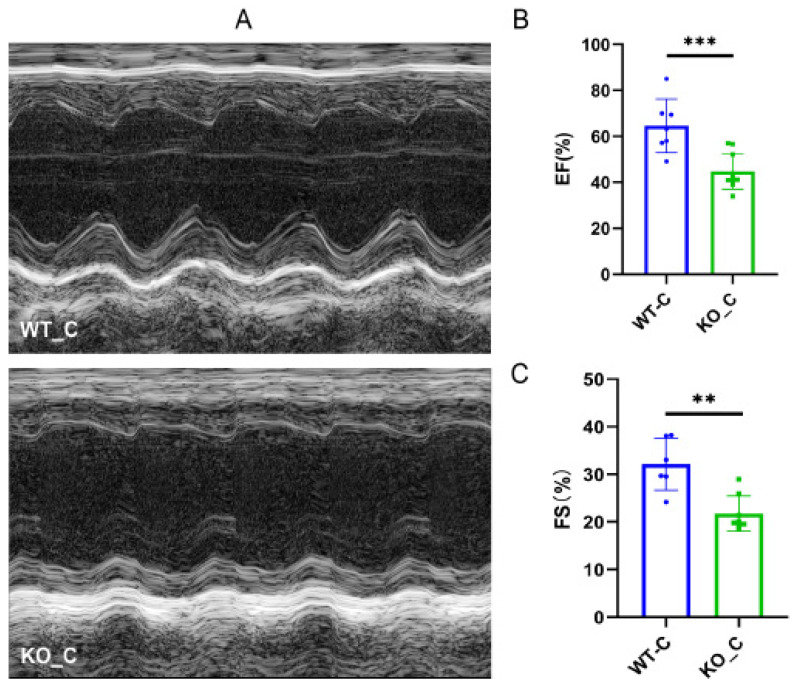
Sigmar1 knockout induced cardiac damage in female mice. (**A**) Representative echocardiography images in M-mode. (**B**,**C**) Indices of cardiac function: EF, FS (%) in mice. *n* = 7–8. Student’s *t*-test was used for statistical analysis. ** *p* < 0.01. *** *p* < 0.001.

**Figure 6 biomedicines-11-02766-f006:**
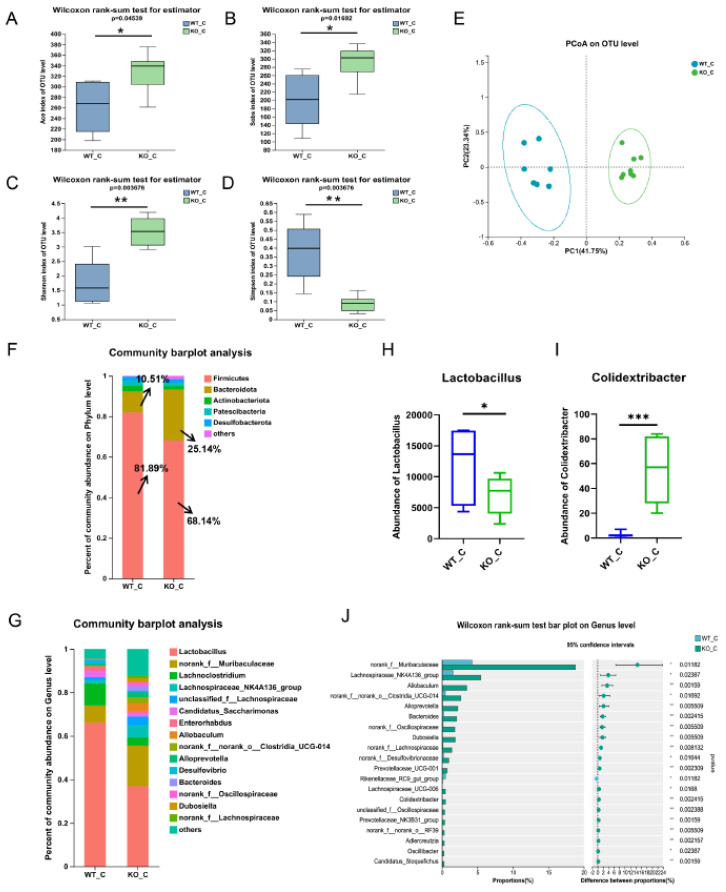
Sigmar1 knockout resulted in alterations in gut microbial composition. (**A**–**D**) α-Diversity indices between the WT_C and KO_C groups. Both the richness and diversity of the gut microbiome were increased in the KO_C group. (**E**) PCoA plot of the gut microbiome. (**F**,**G**) The composition of the gut microbiome at the phylum and genus levels. (**H**–**J**) Significance test of differential bacteria at the genus level using the Wilcoxon rank-sum test followed by the FDR-corrected post hoc test. * *p* < 0.05, ** *p* < 0.01, *** *p* < 0.001.

**Figure 7 biomedicines-11-02766-f007:**
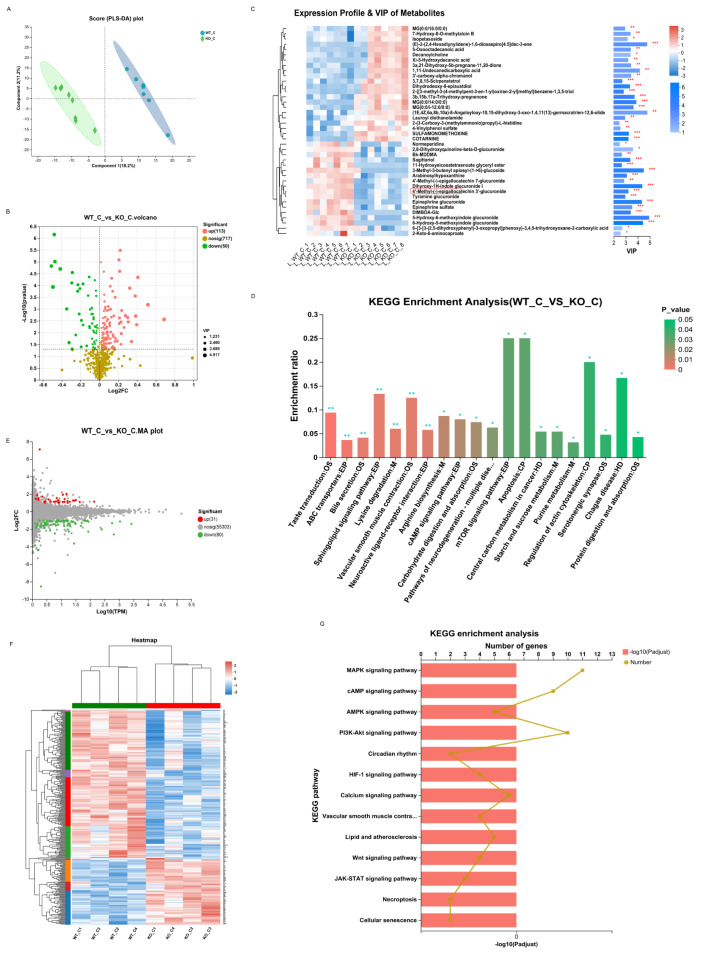
Sigmarr1 knockout resulted in variation in serum metabolites and the transcriptome of the heart. (**A**) Score plot of PLS-DA between the WT_C and KO_C groups. (**B**) Volcano plot of differentially abundant metabolites. (**C**) Heatmap demonstrating the expression profile and VIP value of metabolites. * *p* < 0.05. ** *p* < 0.01. *** *p* < 0.001. (**D**) KEGG enrichment analysis using the hypergeometric distribution algorithm. The mTOR signaling and apoptosis pathways were the most significantly enriched. (**E**) The MA plot of the DEGs. (**F**) The expression of DEGs is shown in the heatmap. There was clear hierarchical clustering of samples between the WT_C and KO_C groups. (**G**) Certain potential pathways were selected for illustration in the bar chart of the KEGG enrichment analysis.

**Figure 8 biomedicines-11-02766-f008:**
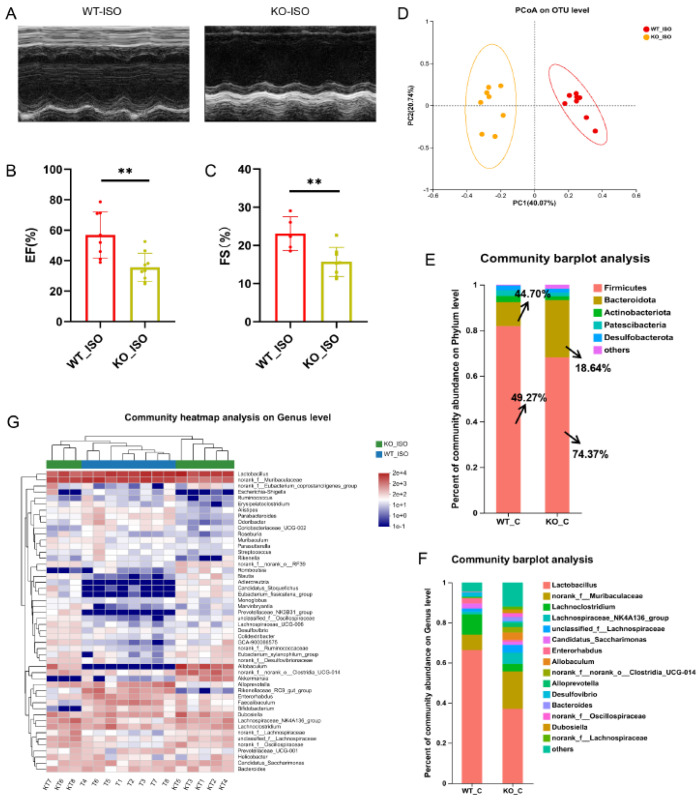
Sigmar1 knockout aggravated cardiac damage in TTS mice and changed the gut microbial community. (**A**) Representative echocardiography images in M-mode. (**B**,**C**) Indices of cardiac function: EF, FS (%) in mice. *n* = 8. ** *p* < 0.01. (**D**) PCoA at the OTU level of the gut microbiome. (**E**,**F**) The composition of the gut microbiome at the phylum and genus levels. There was a noticeable difference in microbial composition. (**G**) Community heatmap analysis illustrates the detailed abundance of bacteria at the genus level.

**Figure 9 biomedicines-11-02766-f009:**
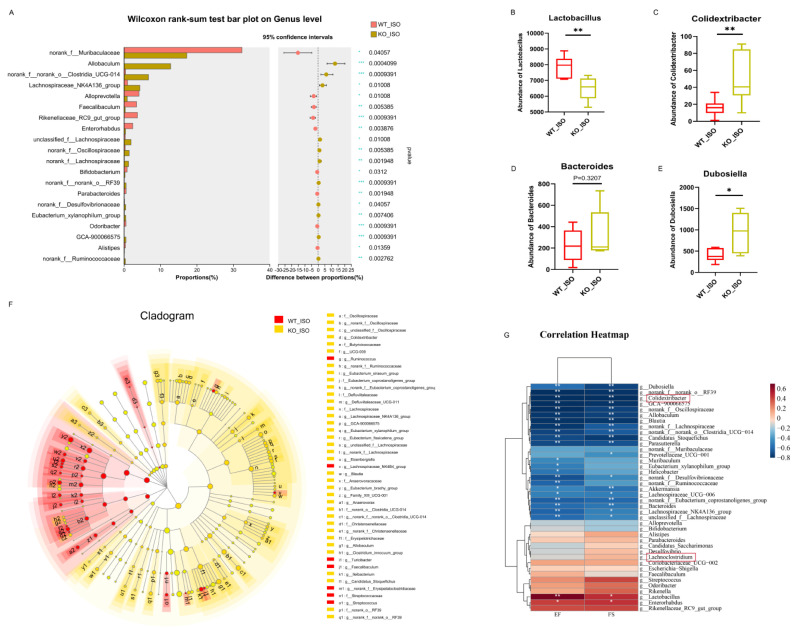
Sigmar1 knockout changed the gut microbial community in TTS mice. (**A**–**E**) Significance test of differential bacteria at the genus level using the Wilcoxon rank-sum test followed by the FDR-corrected post hoc test. * *p* < 0.05, ** *p* < 0.01. *** *p* < 0.001. Lactobacillus, Colidextribacter, Bacteroides, and Dubosiella showed noticeable changes in abundance. (**F**) Cladogram of communities that had significantly different impacts on sample division by LEfSe analysis. (**G**) Correlation heatmap between differential bacteria and indices of cardiac dysfunction.

**Figure 10 biomedicines-11-02766-f010:**
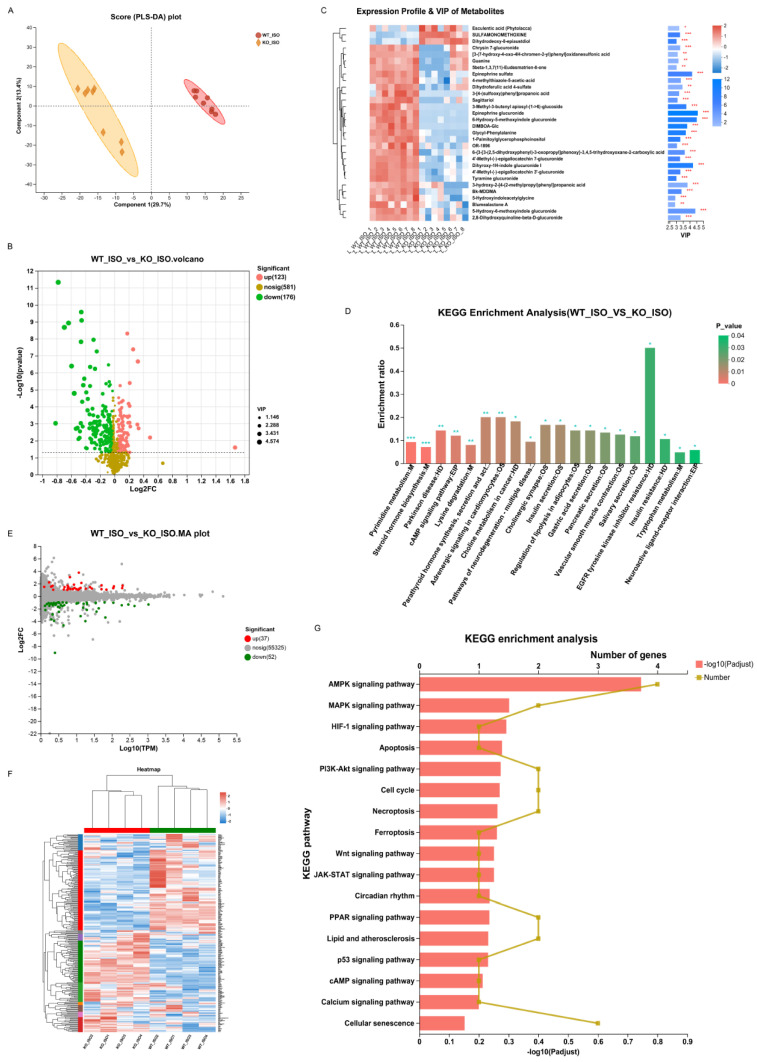
Sigmarr1 knockout altered serum metabolites and the heart transcriptome in TTS mice. (**A**) Score plot of PLS-DA between the WT_ISO and KO_ISO groups. (**B**) Volcano plot of differentially abundant metabolites. (**C**) Heatmap demonstrating the expression profile and VIP values of metabolites. (**D**) KEGG enrichment analysis of differentially abundant metabolites. * *p* < 0.05, ** *p* < 0.01, *** *p* < 0.001. (**E**) The MA plot of the DEGs. (**F**) DEGs are shown in the heatmap. Samples between the WT_ISO and KO_ISO groups were found to cluster well. (**G**) Particular potential KEGG pathways were selected for illustration in the bar chart.

## Data Availability

The raw data supporting our findings have been deposited in the NCBI Sequence Read Archive under accession numbers PRJNA917461 and PRJNA918572, and data will be made available on request.

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
