# Peer review of "Multi-Omics Analysis Reveals the Role of Sigma-1 Receptor in a Takotsubo-like Cardiomyopathy Model"

_biomedicines, 2023, doi:10.3390/biomedicines11102766_

Round 1

Reviewer 1 Report

Manuscript titled “ Multi-omics analysis reveals the role of Sigma-1 receptor in a Takotsubo-like cardiomyopathy model“ is a very interesting article in the field of cardiology and cardiomyophaties.

The overall structure is of good quality and easy to read.

Methods and Results are clear and results corroborate the initial hypothesis of the authors.

Figures and Tables are of sufficient quality and easy to read as well as to understand to readers. However, manuscript need some improvements, specifically in Introduction and/or Discussion. Here the points:

1. In discussion authors should explain the natural bioactives that could improve pathogenesis of Takotsubo syndrome like quercetin and polydatin that are natural inhibitors of IL-6, IL-1 and NLRP3 inflammasome ( key players of Takotsubo) ( cite 10.1007/s10973-017-6135-5 and 10.3389/fonc.2021.680758)

2. Takotsubo could affect cancer patients treated with cardiotoxic drugs, please explain the possible association

3. Authors should explain how takotsubo syndrome could be involved in COVID-19 diseases. Please discuss on this point.

Based on these changes, the article could be suitable for publication in this journal.

Reviewer 2 Report

Dear Editor,

I read this paper and I think that:

- please include more numerical data in the results section of the abstract

- indeed, the role of adrenergic receptors should be discussed in this setting 

Reviewer 3 Report

The authors of the manuscript (MS) “Multi-omics analysis reveals the role of Sigma-1 receptor in a Takotsubo-like cardiomyopathy model” present results of experimental study in mice with and without Sigma1 receptors in whom Takotsubo-like cardiomyopathy was induced pharmacologically with isoprenaline (ISO). The main finding of the study suggests SigmaR1 conveys some protective properties as the extend of cardiac dysfunction (low ejection fraction/fractional shortening) was significantly greater in the knock-out animals. The authors also carried out metabolome and transcriptome analysis, which suggests potential main pathways (among others – HIF1, MAPK,  mTOR and apoptosis) involved in the cardiac dysfunction in this model. In addition, they evaluated changes in the gut microbiome. 

The MS is well written and presents interesting and large set of experimental data. However, there are several flaws that should be addressed. 

1. Based on the introduction, results, and discussion, I am not sure where the link is between Takotsubo-like model of heart failure and gut microbiota. Authors carried out evaluation of microbiome, however, it is not clear how this is related to changes in the cardiac function (echocardiography), metabolome, SimgaR1 to gut bacteria. What is the cause and what is the results, and why gut microbiome?  

2. Statistical analysis – please, confirm in the MS that all data sets are normally distributed and can be evaluated by student t-test. Otherwise, appropriate tests (Mann-Whitney test or other non-parametric tests) should be done and median values with interquartile range should be provided. 

3. More details on mice used in the study should be provided: 

- Were experiments synchronised with oestrous cycle?  

- It is not clear how many mice there are and how many groups were formed. Are there eight wild-type mice and eight SigmaR1 knock-outs divided into control - PBSl (n-4) and experimental - ISO (n-4) groups? I suggest adding a flow chart/protocol scheme.  

- How many hours/days elapsed between administration of ISO and echocardiography/sacrifice of animals? 

4. Figures are small and illegible. 

5. Please, comment if knock-out of SigmaR1 limited only to the heart would affect the results. In other words if the observed protective effect of SigmaR1 is local – cardiac, or systemic. 

6. I suggest adding a recent two-part position statement on pathophysiology of Takotsubo syndrome from the European Society of Cardiology (please, note that I am not associated with the paper nor any of the authors): 

Omerovic E, Citro R, Bossone E, Redfors B, Backs J, Bruns B, Ciccarelli M, Couch LS, Dawson D, Grassi G, Iacoviello M, Parodi G, Schneider B, Templin C, Ghadri JR, Thum T, Chioncel O, Tocchetti CG, van der Velden J, Heymans S, Lyon AR. Pathophysiology of Takotsubo syndrome - a joint scientific statement from the Heart Failure Association Takotsubo Syndrome Study Group and Myocardial Function Working Group of the European Society of Cardiology - Part 1: overview and the central role for catecholamines and sympathetic nervous system. Eur J Heart Fail. 2022 Feb;24(2):257-273. doi: 10.1002/ejhf.2400. Epub 2022 Feb 16. PMID: 34907620. 

Omerovic E, Citro R, Bossone E, Redfors B, Backs J, Bruns B, Ciccarelli M, Couch LS, Dawson D, Grassi G, Iacoviello M, Parodi G, Schneider B, Templin C, Ghadri JR, Thum T, Chioncel O, Tocchetti CG, van der Velden J, Heymans S, Lyon AR. Pathophysiology of Takotsubo syndrome - a joint scientific statement from the Heart Failure Association Takotsubo Syndrome Study Group and Myocardial Function Working Group of the European Society of Cardiology - Part 2: vascular pathophysiology, gender and sex hormones, genetics, chronic cardiovascular problems and clinical implications. Eur J Heart Fail. 2022 Feb;24(2):274-286. doi: 10.1002/ejhf.2368. Epub 2021 Nov 3. PMID: 34655287.

Reviewer 4 Report

This paper investigates the role of the Sigma-1 receptor in isoprenaline-induced Takotsubo-type cardiomyopathy using a murine model. All analyses are supported by strength statistical methods.

One clarification is requested:

Please define more clearly numbers of mice in the 4 groups: WT_C, WT_ISO, KO_C and KO_ISO mice, with a table and a flow chart (in particular, exclusion of one sample in the WT_C group for Isoprenaline-induced serum metabolic disturbances).

1. Introduction :

Line 48: sigma-1 receptor instead of sigma-r receptor.

2. Material and methods:

Line 139: define VIP.

Line 166: define SEM.

Figure 6: *** missing in the legend.

Figure 7: ** missing in the legend.

Round 2

Reviewer 3 Report

Thank you for addressing most of my points.

I still have two issuses related to methods:

1. Please, confirm in the MS, section 2.8 that all data sets were evaluated for normal distribution (please, include test used for this evalution) and fulfill criteria for parametric student t-test (please).

2. Why SEM is used instead of standard deviation? Standard deviation is more meaningful when interpreting figures/data.

3. Please, clarify how many groups there are. Still the sentence "We randomly divided wild-type mice and Sigmar1 knockout mice into two distinct groups (WT_C, WT_ISO, KO_C, KO_ISO, n = 8 in each group)." (L110-111) This sentence is confusing. There are four groups, you get two types of mice and two treatments. Thus, wild-type mice and Sigmar1 knockout mice were randomly assigned to ISO or vehicle forming four groups.

Minor:

The sentence:

"These substances may therefore be considered viable candidates for improving the pathogenesis of TTS" (L509) - what do you mean by "improving the pathogenesis"?
